# Biochemical Properties of Atranorin-Induced Behavioral and Systematic Changes of Laboratory Rats

**DOI:** 10.3390/life12071090

**Published:** 2022-07-20

**Authors:** Patrik Simko, Andrea Leskanicova, Maria Suvakova, Alzbeta Blicharova, Martina Karasova, Michal Goga, Mariana Kolesarova, Bianka Bojkova, Petra Majerova, Nela Zidekova, Ivan Barvik, Andrej Kovac, Terezia Kiskova

**Affiliations:** 1Institute of Biology and Ecology, Faculty of Sciences, Pavol Jozef Safarik University, 040 01 Kosice, Slovakia; patrik.simko1@student.upjs.sk (P.S.); andrea.leskanicova@student.upjs.sk (A.L.); michal.goga@upjs.sk (M.G.); mariana.kolesarova@upjs.sk (M.K.); bianka.bojkova@upjs.sk (B.B.); 2Institute of Chemistry, Faculty of Sciences, Pavol Jozef Safarik University, 040 01 Kosice, Slovakia; maria.suvakova@student.upjs.sk; 3Institute of Pathology, Faculty of Medicine, Pavol Jozef Safarik University, 040 01 Kosice, Slovakia; alzbeta.blicharova@upjs.sk; 4Small Animal Clinic, University of Veterinary Medicine and Pharmacy, 041 81 Kosice, Slovakia; martina.karasova@uvlf.sk; 5Institute of Neuroimmunology, Slovak Academy of Sciences, 831 01 Bratislava, Slovakia; petra.majerova@savba.sk (P.M.); andrej.kovac@savba.sk (A.K.); 6Biomedical Center Martin (BioMed), Jessenius Faculty of Medicine in Martin, Comenius University, 814 99 Bratislava, Slovakia; zidekova27@uniba.sk; 7Institute of Physics, Faculty of Mathematics and Physics, Charles University, 110 00 Prague, Czech Republic; ivan.barvik@mff.cuni.cz

**Keywords:** atranorin, microsomal stability, human serum albumin, behavioral changes, laboratory rats, metabolomics

## Abstract

Atranorin (ATR) is a secondary metabolite of lichens. While previous studies investigated the effects of this substance predominantly in an in vitro environment, in our study we investigated the basic physicochemical properties, the binding affinity to human serum albumin (HSA), basic pharmacokinetics, and, mainly, on the systematic effects of ATR in vivo. Sporadic studies describe its effects during, predominantly, cancer. This project is original in terms of testing the efficacy of ATR on a healthy organism, where we can possibly attribute negative effects directly to ATR and not to the disease. For the experiment, 24 Sprague Dawley rats (Velaz, Únetice, Czech Republic) were used. The animals were divided into four groups. The first group (n = 6) included healthy males as control intact rats (♂INT) and the second group (n = 6) included healthy females as control intact rats (♀INT). Groups three and four (♂ATR/n = 6 and ♀ATR/n = 6) consisted of animals with daily administered ATR (10mg/kg body weight) in an ethanol–water solution *per os* for a one-month period. Our results demonstrate that ATR binds to HSA near the binding site TRP214 and acts on a systemic level. ATR caused mild anemia during the treatment. However, based on the levels of hepatic enzymes in the blood (ALT, ALP, or bilirubin levels), thiobarbituric acid reactive substances (TBARS), or liver histology, no impact on liver was recorded. Significantly increased creatinine and lactate dehydrogenase levels together with increased defecation activity during behavioral testing may indicate the anabolic effect of ATR in skeletal muscles. Interestingly, ATR changed some forms of behavior. ATR at a dose of 10 mg/kg body weight is non-toxic and, therefore, could be used in further research.

## 1. Introduction

Lichens belong to a group of symbiotic organisms composed of green algae or cyanobacteria and fungi. They have been used for centuries in medicine or other industries and they represent a food source for a wide range of animals and humans [1].

Atranorin (ATR), one of the secondary metabolites of lichens, is produced by the “tree moss” also known as *Pseudovernia furfuracea*. It is an aromatic depside that was first isolated from *Stereocaulon alpinum* and shows a wide range of positive properties, which include antibacterial [2], antifungal [3], analgesic and anti-inflammatory [4], cytotoxic [5,6], antiviral [7] and immunomodulatory effects [8]. However, its effect on a healthy living organism has not been comprehensively studied yet. Current studies dedicated to the investigation of this secondary metabolite of lichens have focused mainly on cancer models. The studies mainly focused on the treatment of various tumor diseases with ATR. For example, a study by Harikrishnan et al. (2021) demonstrated its cytotoxic effects in TBARS MB-231 and MCF-7 breast tumor cells in a differential and dose-dependent manner with IC50 concentrations of 5.36 ± 0.85 μM and 7.55 ± 1.2 μM, respectively [9]. In a recent publication by Majchrzak-Celińska et al. (2022) dealing with the supportive treatment of glioblastoma with ATR, a positive effect of ATR on the viability of A-172 at 25 µM dose, T98G at 50 µM dose and U-138 MG at 25 µM dose brain tumor cells was recorded [10]. Zhou et al. (2017) investigated the effect of ATR in vitro on the A549 human lung cancer cell line. They found that at a dose of 10 μg/mL, migration and invasion of these cells were inhibited. They also investigated the effects of ATR in vivo on A549 human lung cancer cells in C57BL/6 mice and observed that ATR at a dose of 5 μg/mL reduced tumor volume, weight, and cell proliferation. Genes such as KITENIN, STAT, c-myc, CD44 and cyclin-D1 were down-regulated, as they interfere with the regulation of the cell cycle, proliferation, and metastasizing [11]. Solar et al. (2016) found that ATR at a concentration of 75 μM significantly reduced the clonogenic ability of 4T1 murine breast cancer cells in vitro. ATR at a dose of 30 mg/kg b. w., administered *per os* over a period of 10 days, decreased the concentration of thiobarbituric acid reactive substances (TBARS) in the liver of BALB/c mice. Therefore, the authors concluded that ATR is an antioxidant with hepatoprotective effects in vivo [12]. ATR and its decomposition particle methyl haematomate are not only a good antioxidants, but also potent immunomodulators (3.1–100 µg/mL) [8]. These and other studies dedicated to investigation of the effect of ATR on tumor cells in vitro, suggest its potential human use in the future.

The study aimed at basic physicochemical properties, the binding affinity to human serum albumin (HSA), basic pharmacokinetics, and, mainly, on the systematic effects of ATR in vivo.

Our results show that ATR is an unstable molecule, especially at acidic and neutral pH, and binds to HSA with a high affinity near the binding site TRP214. Moreover, during the treatment of healthy animals, ATR at the dose of 10 mg/kg body weight influences some forms of behavior, causes mild anemia and has no impact on liver. These aspects should be considered when designing the further experiments dedicated to investigating the impact of ATR during various diseases.

## 2. Materials and Methods

### 2.1. Lichen Material and Preparation of Lichen Extract

Lichen *P. furfuracea* (L.) Zopf. was collected randomly from the branches of pines and spruces in Špania dolina (48°49′ N, 19°08′ E) central Slovakia, 730 m a. s. l. The collected material was washed in distilled water and then left at room temperature (23 °C) for 48 h to air-dry. The lichen material was rinsed with distilled water and air-dried at room temperature (26 °C) for 48 h. A total of 20 g/DW of *P. furfuracea* was homogenized using liquid nitrogen in mortar. The pounded powder was then placed in Erlenmayer flasks and secondary metabolites were isolated using acetone (50 mL/5 g DW) for 24 h. The collected extract was filtered using a nylon sifter into a Petri dish and was left to evaporate under laboratory conditions. The crystals of secondary metabolites were then scraped and collected.

### 2.2. Identification of Lichen Substances by High-Performance Liquid Chromatography (HPLC) Analysis and Nuclear Magnetic Resonance (NMR) Spectroscopy

The powder from the extraction procedure was analyzed by semi-preparative HPLC with DAD detection (Agilent Technologies 1260 Infinity device, Waldbronn, Germany). A 7 µm Kromasil SGX C18 column (Merck Life Science, Bratislava, Slovakia) was used. Mobile phase A (5% acetonitrile + 1% (*v*/*v*) trifluoracetic acid) and mobile phase B (80% acetonitrile) were in isocratic program with a flow rate of 0.7 mL min^−1^: 0 min 50% A and 50% B; 25 min 0% A and 100% B; 30 min 50% A and 50% B. For quantitative analysis of GA, the wavelength of 254 nm was used.

NMR spectra were recorded at room temperature (25 °C) on a Varian VNMRS spectrometer (IET|International Equipment Trading Ltd., Mundelein, IL, USA) operating at 599.87 MHz for 1H and 150.84 MHz for 13C. Spectra were recorded in CDCl3-d1 and the chemical shifts were referenced to the internal reference standard TMS (1H NMR 0.00 ppm, 13C NMR 0.00 ppm). The 2D gHSQC and gHMBC (optimized for a long-range coupling of 8 Hz) methods were employed.

### 2.3. Molecular Modeling of Atranorin

The molecular model of ATR was built using the Molefacture module from the VMD software package (Software version 1.9.4, Board of Trustees, University of Illinois, Champaign, IL, USA) [13]. Docking calculations were carried out using AutoDock Vina version 1.1.2 program [14], and the UCSF Chimera graphical interface [15]. The results were visualized and figures were produced using the UCSF Chimera program [15].

### 2.4. pH Properties and Stability

The stock solution of 2 mM ATR was prepared in 96% ethanol and stored at 8 °C in the dark. The working solution of ATR was prepared by the mixing of 5 μL stock solution in 2000 μL of distilled water. The stability of ATR water solution at acidic, neutral, and alkaline pH was monitored by repeated UV–Vis spectral measurements in time. pH was measured by Hanna Instruments HI 2211 pH/ORP Meter (Hanna Instruments, Norfolk, VA USA) using Hanna Instruments HI 1131 B electrode. UV–Vis spectra were obtained by Specord S300 UV–Vis spectrometer (Analytik Jena AG, Jena, Germany) in wavelength range 184.5 to 550 nm. Quartz cuvette with 1 cm light path was used. pH properties were studied as pH-dependent changes in UV–Vis spectra of continuously stirred ATR water solution. The concentration of ATR in the working solution was 5 μM and spectra were measured at 22 °C. The ATR water solution was alkalized using 4 M NaOH continuously until pH 11. Data obtained for λ 239, 308 and 332 nm were analyzed by GraphPad Prism 8.01 software (GraphPad Software, Inc., San Diego, CA, USA) and pH curves were fitted using the sigmoidal standard curve (Sigmoidal, 4PL, X is log (Concentration)). Experimentally obtained data were compared to values of ATR pKa calculated by Chemicalize.com (accessed on 16 May 2022) web service provided by © 2022 ChemAxon (Budapest, Hungary).

### 2.5. Binding to Human Serum Albumin

ATR was dissolved in DMSO (CAS no.: 67-68-5, ≥99.8% purity, Sigma Aldrich, St. Louis, MO, USA) to obtain a stock solution with a concentration 0.7 M. Stock solution was stored at 20 °C in the dark. Albumin from human serum (A1887, CAS no.: 70024-90-7) and reagents for buffer solutions were obtained from Sigma Aldrich, St. Louis, MO, USA.

Human serum albumin (HSA) stock solution was prepared by dissolving 20 g/mL in 150 × 10^−3^ M Tris-HCl-NaCl (pH 7.4, 22 °C). Concentration of stock solution was determined by UV–Vis spectroscopy (Specord S300 UV VIS, Analytik Jena, Jena, Germany) using quartz cuvette with 1 cm optical length. Extinction coefficient of HSA used in the calculation was ε280 = 35,700 M^−1^cm^−1^ [16]. HSA stock solution was stored at 4–8 °C in the dark.

The fluorescence quenching spectra were obtained by Varian Cary Eclipse spectrofluorimeter (Varian Medical Systems, Inc., Palo Alto, CA, USA) in the wavelength range 290–500 nm using a quartz cuvette with an optical length of 1 cm. The excitation slit width was 5 nm and the emission slit width was set at 10 nm. Excitation wavelength 280 nm, averaging time 0.5 s and 1 nm data interval were used. Spectra were measured at fixed HSA concentration (c = 4.0 µM) and increasing concentrations of ATR (0.4–3.5 µM) in 10 mM Tris-HCl buffer solution (pH 7.4, 22 °C). Measurements were performed at 15 °C, 25 °C and 37 °C.

Synchronous fluorescence spectra were recorded at two different scanning intervals ∆λ, while ∆λ = λem − λex. The excitation wavelength was set in the range of 200–350 nm and the fluorescence emission was obtained at ∆ = 15 nm for tyrosine (Tlos 200yr) residues and ∆ = 60 nm for tryptophan (Trp) residues. The concentrations of HSA and the ligands were the same as in the fluorescence spectra measurements. The temperature of all measurements was 25 °C.

The ATR molecule was docked to the binding site of HSA, which is occupied by Thyroxine in the original crystal structure (PDB id: 1HK1) [17].

### 2.6. Microsomal Stability

The method described by Di et al. [18] for microsomal stability assay of insoluble compounds was used. Briefly, the compounds were solubilized in DMSO (CAS no.: 67-68-5, ≥99.8% purity, Sigma Aldrich). Concentrated rat liver microsomes (ThermoFischer Scientific, Bratislava, Slovakia) were mixed with pre-warmed phosphate buffer (100 mM) and the compounds. The reaction was initiated with the addition of 20 mM NADPH (CAS no: 2646-71-1Merck Life Science, Bratislava, Slovakia). The mixture was incubated at 37 °C for up to 120 min, terminated by the addition of acetonitrile and centrifuged (30,000× *g*/10 min). The supernatant was transferred into vials and analyzed with LC-MS/MS.

### 2.7. Ultra-Performance Liquid Chromatography Coupled to Tandem Mass Spectrometry

A Waters triple-quadrupole mass spectrometer (Waters Corporation, Prague, Czech Republic)with an electrospray ionization source in negative mode (ESI-) was used. The mass spectrometer was operated with the following parameters: capillary voltage 3 kV, source temperature 150 °C, desolvation temperature 350 °C, cone gas 50 L/h, and desolvation gas 800 L/h. The source cone voltages and collision energies were manually optimized for each SRM transition. The chromatographic apparatus consisted of a Waters ACQUITY UPLC system with a binary gradient pump, autosampler and column thermostat (Waters, Prague, Czech Republic). Chromatographic separation was performed on an ACQUITY UPLC BEH C18 (2.1 × 50 mm, 1.7 μm particle size) column. The column temperature was maintained at 30 °C. Mobile phase A consisted of 0.1% formic acid in water. Mobile phase B consisted of 100% acetonitrile. The elution started at 20% B (0–0.5 min), increasing to 90% B (0.5–2 min) to 90% B (2–3 min), returning to 20% B and re-equilibrating at 3–4 min. The flow rate was 0.5 mL/min, and the injection volume was 5 μL. MassLynx software version 4.1 (Waters Corporation, Prague, Czech Republic) was used for instrument control and data acquisition and analysis.

### 2.8. Experimental Animals

Twenty-four Sprague Dawley rats (Velaz, Únetice, Prague, Czech Republic) were used in the experiment. The animals were kept under standard conditions with a room temperature ranging from 21 to 24 °C with a relative humidity of 50–65% and a 12:12 h light:dark regimen. The animals were provided with tap water and pelleted food (Velaz, Únetice, Prague, Czech Republic) ad libitum according to EU animal-fed legislation and guidance. The animals were handled by the guidelines established by Law No. 377 and 436/2012 of Slovak Republic for the Care and Use of Laboratory Animals (Ro-2866/16-221, Ro-2219/19-221/3). The animals were divided into four groups. The first group (n = 6) included healthy males as intact control (♂INT) and the second group (n = 6) included healthy females as intact control (♀INT). Another two groups (♂ATR/n = 6 and ♀ATR/n = 6) consisted of animals with daily administered ATR (10 mg/kg body weight) dissolved in 10% ethanol solution *per os* for a one-month period, starting at the age of 2 months. At the age of 2 and 3 months, the blood was taken. Blood samples were collected from the tail vein for biochemical and serum analyses. Samples with whole blood in K_3_EDTA tubes were analyzed by using an automated veterinary hematology analyzer (MindrayBC 2800VET, Shenzhen, China). Serum samples were stored at 4 °C and biochemistry analysis was performed over two consecutive days using an automated clinical chemistry analyzer (Cobas c 111 analyzer, Roche, Basel, Switzerland) according to the manufacturer’s instructions. At the end of the experiment, the liver was excised *post mortem*. The histology was performed. TBARS were measured according to [19].

### 2.9. Behavioral Analyses

All experimental rats aged 3 months were subjected to behavioral testing to evaluate anxiety and exploratory behavior in these animals. We chose the elevated plus maze (EPM), and the open field test (OFT) as behavioral tests.

The OFT consisted of a white box sized 100 × 100 cm with wooden walls that were 60 cm in height. In OFT, the locomotor activity—total traveled distance, an average speed of each animal, time spent on the periphery, and time spent in the center—were measured. We also observed exploratory behavior in terms of rearing, defecating and comfort behavior when washing. At the start of the test, the rat was placed in the center of the field. Each trial lasted 6 min. The testing progress of the animals in the open field test was video-recorded and evaluated using the computerized video-tracking system Smart Junior (Panlab, Barcelona, Spain).

For the EPM, a cross labyrinth with two open and two closed arms was used, with an arm length of 80 cm, between which was a central square of 15 × 15 cm. The arms of the maze were at a height of 75 cm above the ground. In this test, we observed and recorded anxiety level—the time spent in the open arms and the frequency of defecation; exploratory activity—the amount of rearing; comfort behavior—the number of washing acts; and locomotor activity—the number of passes through the center of the maze. At the start of the test, the rat was placed into the center of the field. Each trial lasted 6 min.

### 2.10. Metabolomics

The blood from all experimental animals was collected at one-time point from the tail vein in a total volume of 100 μL into microtubes. The place of the collection was treated with a disinfectant. After isolating, blood serum was stored at −80 °C. Frozen serum was thawed on ice and used for further analysis. The profiling of metabolites was done by liquid chromatography-tandem mass spectrometry (LC-MS/MS) using the Biocrates AbsoluteIDQ p180 kit (Biocrates Life Sciences AG, Innsbruck, Austria). The kit enables simultaneous determination and quantification of 185 metabolites from six classes (amino acids, biogenic amines, acylcarnitines, glycerophospholipids—consisting of lysophosphatidylcholines and phosphatidylcholines, sphingolipids, and a sum of hexoses—including glucose). Chromatographic separation was achieved using the ultra-performance liquid chromatography system ACQUITY UPLC I-Class (Waters, Prague, Czech Republic) coupled with triple quadrupole mass spectrometer XEVO TQ-S (Waters, Prague, Czech Republic). Metabolites were extracted and measured following the manufacturer’s instructions as described previously in studies [20,21].

Briefly, after adding internal standards on the filter spot of the 96-well extraction plate, 10 μL of samples, quality control (QC) samples or calibration standards were pipetted into appropriate wells, respectively. Subsequently, amino acids and biogenic amines were derivatized with phenyl isothiocyanate (CAS No: 98 103-72-0, Sigma-Aldrich, Steinheim, Germany) and dried. During the next step, metabolites were extracted with ammonium acetate in methanol (CAS No: 99.8 67-56-1, Sigma-Aldrich). Finally, the extracts were appropriately diluted and analyzed in two separate analyses. Metabolites belonging to amino acid and biogenic amine classes were analyzed by the LC-MS/MS part of the method, and analytes were separated on the ACQUITY UPLC BEH C18 (2.1 × 75 mm, 1.7 μm). Flow injection analysis (FIA)-MS/MS part of the method was used for the determination of metabolites belonging to acylcarnitines, glycerophospholipids, sphingolipids and hexose classes. Metabolites from LC-MS/MS part were quantified by isotopic dilution on a 7-point calibration curve, and metabolites from FIA-MS/MS were quantified by their relative intensities over the chosen isotopically labeled internal standards (semiquantitative approach).

Metabolomics measurement of a selected group of lysoPCs, PCs and SMs was performed on serum samples. The fully automated assay was based on PITC (phenylisothiocyanate) derivatization in the presence of internal standards followed by FIA-MS/MS and LC-MS/MS using a SCIEX 4000 QTRAP^®^ (SCIEX, Darmstadt, Germany) or a Waters XEVO™ TQMS (Waters, Vienna, Austria) instrument with electrospray ionization. The assay was based on the principle described in the study of Pena et al. [22]. The values were log2-transformed to obtain normally distributed data and to stabilize the variance.

### 2.11. Statistical Analysis

All statistical analyses were performed using GraphPad Prism 8.0.1 (GraphPad Software, Inc., San Diego, CA, USA). All data were examined for normal distribution, and appropriate tests were applied. The data are presented as mean ± standard deviation (SD).

Quantification of metabolite concentrations and quality assessment were performed using the MetIQ software package (BIOCRATES Life Sciences AG, Innsbruck, Austria). Internal standards served as the reference for the metabolite concentration calculations. Univariate (*t*-tests and ANOVA) and multivariate statistics (partial least squares-discrimination analysis PLS-DA) as well as the variable importance in projection (VIP) plot, were performed using MetaboAnalyst 5.0 (Xia Lab @ McGill, St. Anne de Bellevue, QC, Canada). Cross-validation of PLS-DA classification applied 5 components for selection as the optimal number of components. The LOOCV cross-validation method was used. It has been considered the performance of measures as Accuracy, R2, Q2. As a part of the PLS-DA method, a VIP score was measured. VIP score is a measure of a feature’s importance in the PLS-DA model. It summarizes the contribution a feature makes to the model. The VIP score of a feature is calculated as a weighted sum of the PLS weight. PLS weight is the squared correlation between the PLS-DA components and the original feature.

## 3. Results

### 3.1. Identification of Lichen Extract by HPLC and NMR

The HPLC chromatogram of the acetone extract of *P. furfuracea* revealed five fractions of lichen secondary metabolites, already published previously [23]. Based on the internal standards, ATR, chloratranorin (as a precursor of ATR), physodalic acid, 3-hydroxyphysodic and physodic acid were identified. For our experiments, ATR was used.

### 3.2. 3D Modeling of Atranorin Structure

Using the Molefacture module, 3D ATR model was performed (Figure 1).

### 3.3. pH Properties and Stability

ATR stability evaluation in a water solution at laboratory temperature from the range of pH 1 to 11 show, at various pH, that it is a highly unstable substance, especially at acidic and neutral pH (data not shown). Therefore, proper pH dependence measurements were not possible. Calculated values of ATR pKa suggested major changes in ATR microspecies distribution at pH greater than 8.

### 3.4. Binding to Human Serum Albumin

To investigate the binding of ATR to HSA, the fluorescence emission spectra of HSA in the presence of ATR in selected concentrations were recorded. In the present case, a linear Stern–Volmer plot was observed for ATR, which means that only one type of quenching mechanism occurs (static or dynamic) [24]. The value of kq 4.679 × 10^13^ L mol^−1^.s^−1^ is greater than the maximum scatter collisional quenching constant given as 2 × 10^10^ M^−1^.s^−1^ [25], which suggests that the fluorescence quenching of HSA by ATR is initiated by static mechanism [24]. The binding constant of ATR (4.258 × 10^6^ dm^3^.mol^−1^) is greater than the optimal value by one order. The optimal value of the binding constant, to ensure that a significant amount of compound gets transported and distributed through the organism, and, at the same time, that the compound can be released once it reaches its target, is considered to be 10^4^–10^6^ M^−1^. The value of n is approximately equal to 1, which indicates that only one independent class of binding site is available for the studied derivative on HSA [26]. Synchronous spectra showed no shift in maximum wavelength observed for Trp and Tyr residues, which suggests that their microenvironment polarity remains unaffected by the binding of ATR to HSA. The curve of ∆λ = 60 nm is lower than the curve of ∆λ = 15 nm for ATR, which suggests that Trp plays an important role during fluorescence quenching of HSA [27]. The same was confirmed by the excitation of HSA in the presence of ATR at 295 nm [28]. The standard Gibbs free energy of HSA–ATR interaction has a negative value, which indicates the spontaneity of the process. On the other hand, positive values of the standard enthalpy and entropy, together with changes in the quenching constant with a temperature indicator, show that a hydrophobic interaction plays the main role in HSA–ATR complex formation [29,30]. The 3D fluorescence spectra can not only reflect the fluorescence information of the protein, but also display the variation of HSA conformation induced by drugs [31]. HSA possessed two fluorescent peaks. Peak 1 (λex/em = 280/342 nm/nm) mainly reveals the spectral behavior of Trp or Tyr residue of HSA (fluorescence of Phe residues can be negligible) (Table 1), whereas fluorescence peak 2 (λex/em = 230/332 nm/nm) exhibits the spectral behavior of polypeptide backbone structures [32]. The fluorescence intensities of the two peaks were reduced under the addition of ATR. The intensity of peak 1 decreased from 192 to 142 with a 6 nm hypsochromic shift, suggesting the ATR interaction with Tyr and Trp residues of HSA. As referred to in peak 2, the fluorescence intensity reduced from 295 to 171 and a 3 nm Stokes shift was observed. These results implied that ATR changed polypeptide backbone conformation causing the alteration in the secondary structure of HSA. The parameters of the calculated quenching and binding parameters of the ATR–HSA system are listed in Table 2. According to docking analysis, ATR binds to HSA near the binding site TRP214 (Figure 2).

### 3.5. Metabolic Stability

We used a rat liver microsome to determine the metabolic stability of ATR. First, we optimized LC-MS/MS parameters to achieve optimal chromatographic separation and sensitivity of the mass spectrometry detection. The spectra indicated that the analyte form protonated molecular ions [M-H+]–*m*/*z* 373. The MS/MS spectra of ATR contained two fragment ions at *m*/*z* 163 and 174, that were used as quantification and qualification transitions. The retention time for targeted analyte was 1.8 min (Figure 3). ATR was unstable in rat liver microsomes with a half-life of 7.6 min.

### 3.6. Body Mass Gain of Laboratory Animals—Food and Water Intake

ATR significantly decreased body mass gain during some weeks (Figure 4A,B). Concomitantly, the food and water intake were not influenced (Table 3).

### 3.7. Behavioral Analysis

General behavior reflects the health status of laboratory animals [33]. Therefore, we tested the animals in standardly used behavioral tests, such as OFT or EPM.

As shown in Table 4, in OFT, rearing activity has been observed at an increased rate in both ATR males and females (*p* < 0.01 or *p* < 0.001, respectively). The number of defecation boluses increased after ATR administration in both sexes compared to INT groups (*p* < 0.05 or *p* < 0.01, respectively). No changes in washing activity, times spent in the center or periphery, traveled distance or average speed were seen.

In the EPM, the number of crossings of the center increased statistically after ATR administration (*p* < 0.05) (Table 5). In addition, the time in open arms has been markedly prolonged in ATR groups (*p* < 0.001 and *p* < 0.05, respectively). Similarly, to OFT the number of defecation boluses increased after ATR administration in both sexes compared to INT groups (*p* < 0.01, respectively). However, the number of rearing in ATR groups stayed unchanged compared to healthy INT rats. Although, this result was not confirmed in OFT.

### 3.8. The Analysis of Blood and Liver in Rats

Blood test data have always been traditionally confined to the clinic for diagnostic purposes, and are still routinely used as a physiological profiling and monitoring tool [34].

#### 3.8.1. Hematological Parameters

Results of the hematological evaluation of the intact control group compared to groups with ATR administration are presented in Table 6. The data show that leukocyte and lymphocyte values were significantly elevated in the male group with ATR administration compared to intact males (*p* < 0.01 and *p* < 0.01, respectively). In the same group with ATR administration, the number of red blood cells was decreased (*p* < 0.05) followed by statistical elevation of some red cell indices; mean corpuscular volume (MCV; *p* < 0.001), mean corpuscular hemoglobin (MCH; *p* < 0.01) and red cell distribution width (RDW; *p* < 0.001). There were no significant differences in the parameters examined in the control and treated groups of females.

#### 3.8.2. Biochemical Parameters

ATR administration increased LDH and creatinine in males (*p* < 0.001 and *p* < 0.05, respectively) and females (*p* < 0.01). There were no significant differences in other biochemical parameters (Table 7).

#### 3.8.3. TBARS in Liver

TBARS (g/tissue) as well as TBARS content in liver, stayed unchanged after ATR administration (Table 8).

#### 3.8.4. Histology of Liver

As shown in Figure 5, no histological abnormalities have been found in ATR groups.

### 3.9. Blood Metabolomics

Sex differences in metabolism/metabolomics have been described before [35,36]. Thus, we decided to analyze separately males and females. As shown in Figure 6, there are predominantly sex differences observed. While females show elevated levels of the 15 most important PCs (Figure 6b) predominantly with an overlapping pattern (Figure 6a—red and green color), males were influenced by ATR treatment (blue and violet color) differently.

We further analyzed the data sex-independently (Figure 7). Partial least discrimination analysis showed that animals treated with ATR have some features in common with healthy animals; however, some of them are changed. As shown in Figure 7, according to the VIP score, the 10 most important features were set. All PCs (PC aa C32:1, PC aa C36:4, PC aa C34:4, PC ae C38:3, PC aa C 38:4 and PC ae C38:4) were decreased in the blood of ATR animals. Acylcarnitine C4, amino acid glutamine (Glu), and biogenic amine spermidine increased in ATR groups. One biogenic amine Carnosine has been found to be decreased in the blood of ATR animals.

## 4. Discussion

The information boom on the biological effects of ATR started several years ago after its perspective was uncovered in research. Nowadays, many authors focus on its antioxidant properties [37,38,39], and/or mainly on its cytotoxic effects predominantly in vitro [5,6,9,23], occasionally in vivo [11,12,40]. We divided our work into two main parts—one in vitro and one in vivo. In our study, we addressed the basic physicochemical properties, binding affinity to HSA, basic pharmacokinetics, and, most importantly, the systematic effects of ATR in vivo. ATR caused mild anemia during treatment. However, based on blood liver enzyme levels (ALT, ALP, or bilirubin levels), TBARS content or liver histology, the liver was not influenced. Significantly elevated Crea and LDH levels, together with increased defecation activity during behavioral testing, may indicate the anabolic effect of ATR in skeletal muscles. Interestingly, ATR altered some forms of behavior.

The stability of a tested compound belongs to one of the basic test types. ATR has been previously described as a temporally unstable molecule when in solution [41]. Our results are in agreement with other studies. The compound is soluble in 96% ethanol and, interestingly, slightly soluble in aqueous solution; however, its stability is strongly dependent on pH. In the monitored range of pH 1 to 11, we observed changes in the absorption spectra of ATR. At acidic and neutral pH in aqueous solution, the ATR molecule signal disappeared rapidly. Conversely, at basic pH, the signal remained stable for a long time. This suggests that in an aqueous environment at acidic and neutral pH, the ATR molecule is in a form that is readily degradable. The ATR form present at strongly basic pH is more stable in aqueous media. ATR was most stable in acetonitrile solvent [41]. It was only partially soluble in other solutions, such as acetone, chloroform, or methanol [41,42]. It was described that ATR decomposed in a methanol solution by transesterification and hydrolysis into atraric acid and methyl haemotommate [41,42,43,44,45]. Various biological effects have also been attributed to atraric acid itself [46,47].

The metabolic stability of ATR was tested using rat liver microsomes; however, due to its instability and relatively difficult solubility, our results could not be presented properly. ATR was unstable in rat liver microsomes with a half-life of 7.6 min. When compared to other secondary metabolites, there are only sporadic reports so far. Some experiments on rabbits indicate that the mean terminal half-life of usnic acid level in plasma was 10–19 h after oral administration [48,49]. Other naturally occurring agents, such as resveratrol, found in red grapes and red wine, have a half-life in blood of around 9.2 h [50]. However, because it undergoes rapid metabolism in the body, parental resveratrol is found in the blood for only a few minutes [51]. Natural substances metabolize in the body predominantly into glucuronides and sulfates. These have their biological activities, sometimes higher than the parental compound, such as resveratrol glucuronide and sulfate [52], genistein glucuronide and sulfate [53], and many others.

Once the xenobiotic gets inside the body, it binds to and is transported by HSA, the most abundant protein in human blood plasma, which constitutes about half of the serum protein and is responsible for the transport of hormones, fatty acids, and other compounds [54]. The knowledge of interaction mechanisms between ATR and HSA is of great importance in the understanding of the pharmacokinetics and pharmacodynamics of this bioactive substance [55,56]. Amino acid residues, tyrosine (Tyr), tryptophan (Trp) and phenylalanine (Phe) are responsible for the fluorescent properties of HSA [57,58]. When HSA interacts with a ligand there are changes in the fluorescence spectra of HSA. Based on these changes, it is possible to describe the nature of the interaction of xenobiotics with HSA. Our results suggest that spontaneous hydrophobic interactions play a major role in the formation of the HSA–ATR complex. Our results also show that ATR has a relatively high affinity for HSA. This may result in a longer persistence of ATR in the blood plasma and a slower release to the sites of action. This suggests that the dosage of ATR administered to a living organism should be chosen carefully [59]. ATR is likely to bind to a single HSA binding site as the data from docking analysis indicate. ATR binds near the TRP214 binding site in HSA. The synchronous spectra of the interaction of ATR with HSA suggest that there are no changes in microenvironment polarity near the Trp and Tyr residues. However, 3D fluorescence spectra show the opposite. The discrepancy between the results may be due to the nature of the 3D measurement. Due to the long measurement, there could be changes in the ATR structure in the aquatic environment and this could affect the measurement results.

We must first understand the effects of a substance on a healthy organism before investigating it as a treatment for a variety of disorders. Therefore, we decided to test ATR in both healthy male and female Sprague Dawley rats. As the data from in vivo studies are rare, in our study, we decided to use ATR in the dose of 10 mg/kg body weight.

Tracking the whole body weight and/or body mass gain is a useful indicator of the health status of the organism [60]. Body mass gain decreased significantly during some of the experimental weeks, concretely during the weeks 9 and 10. From the beginning of the 9th experimental week, ATR was administered daily to the animals. As we can see, the first two weeks after the administration, the body mass gain decreased significantly in both sexes. However, at the end of the experiment, the body mass gain was not changed when compared to untreated INT rats. Thus, we wanted to see if the concomitant decrease could be seen also in food and water intake. But neither food nor water intake was influenced during ATR administration. Concomitantly with these results, the markedly increased defecation has been detected during behavioral testing in both EPM and OFT. We also measured significantly elevated LDH and Crea levels in both ATR groups.

Elevated LDH levels in the ATR male and ATR female group could indicate liver tis-sue or muscle damage. In our study, there were no elevated ALT, ALP and bilirubin levels; thus, liver function remained unchanged or may also have stabilized by the time ATR administration was discontinued. Concomitantly, no histological abnormalities in the liver were seen. TBARS content stayed unchanged after ATR administration. At the same time, however, the impact of ATR on liver function cannot be excluded, as in a study on the acute (5 g/kg single dose) and subchronic toxicity (30 mg/kg for 30 days) of ATR, significantly elevated bilirubin and ALP values were found in some groups, which may indicate biliary cholestasis and reduced liver function [23]. In the aforementioned study, although the LDH level decreased slightly after ATR application, there was still an increase in LDH in the ATR group of females after a dose of 5 g/kg. The amount of Crea that is formed daily in the body is influenced by the rate of its synthesis and especially by the total mass of skeletal muscle. Elevated Crea levels can be caused by feeding with a high-protein diet [61]. ATR rats and intact rats in our study were fed in the same way; therefore, the effect of dietary protein on Crea elevation in the ATR female and male group is excluded. Serum urea and Crea concentrations are markers of excretory renal dysfunction, but their concentrations rise in parallel only when at least 75% of renal filtration capacity is lost [62]. Since urea values did not change in our rats and the animals did not show symptoms of azotemia, we do not assume that glomerular damage and a decrease in renal glomerular filtration capacity occurred. However, the urea cycle was partly disturbed by the elevated spermidine, Crea and Glu levels. Thus, some effects on muscle fibers can be inferred from the elevated LDH, and Crea values in the ATR groups. In rats treated with ATR, blood carnosine levels, C4 acylcarnitine, and several PCs were reduced and more frequent defecation, with no change in the volume of feces excreted per day, was recorded, which may indicate increased metabolic activity. Thus, we hypothesize that administration of ATR may induce an anabolic effect in skeletal muscle, and possibly local irritation of muscle fibers.

Hematological analysis of blood parameters revealed some interesting points. Firstly, the leukocytes and lymphocytes in males were elevated together with decreased RBCs and elevated MCV, MCH and RDW. All these changes refer to dehydration or a mild anemia [63]. Because we saw the increased defecation pattern during behavioral testing, we cannot exclude dehydration as a point leading to the anemia of experimental animals. Secondly, in females no changes in hematological parameters after ATR administration were recorded. When seeing the data from the metabolomics analysis, there were also indices that ♀ATR and ♀INT show overlapping patterns during PLS-DA. Such an interesting point leads us to assume that there could be some protective mechanism in the female body that restricts ATR to manifest. It is known that in women, endogenous female sex hormones, especially estrogens, are cardioprotective via multiple mechanisms: increased high-density lipoprotein, decreased low-density lipoprotein, and release of vasodilators, such as nitric oxide, and prostacyclin PGI_2_ from the vessel walls, which result in an inhibition of vascular constriction and a lowering of blood pressure [64]. It is not surprising that the adverse drug reactions have been described as gender specific [65,66]. Sex differences have also been previously described during the treatment with other natural substances, such as resveratrol [67,68,69], melatonin [70], epigallocatechin gallate [71] or silymarin [72].

Another interesting point of our experiment was revealing the behavioral changes after ATR consumption. In OFT, (except for the defecation boluses described before) the rearing activity increased significantly in both ATR groups. Other parameters (washing, traveled distance, and average speed, time spent in the center and periphery) stayed unchanged. In EPM on the other hand, rearing was not influenced by ATR, but, except for statistically elevated defecation boluses already mentioned before, center crossings together with the time spent in open arms were markedly increased. Our results indicate that the anxiety level of the animals (expressed as elevated time in open arms and center crossings in EPM) after ATR consumption decreased when compared to INT animals. In general, the increased time spent in open arms indicates a lower degree of “anxiety” in the animals [73,74]. On the other hand, in the exploratory test, the rearing activity of ATR animals increased. Exploratory rearing is one of the main forms of exploration of a new place [75]. All these manifestations in behavior may suggest that ATR could have some antidepressant potential. The patients with depressive disorders or bipolar affective disorder (manic depression) are often without interest in the outside world and their exploration activity is reduced [76]. People with symptoms of depression show impairments in decision-making. One explanation is that they have difficulty maintaining rich representations of the task environment [77].

Our results offer a complex overview of ATR attributes that could be used in further research. ATR is not a stable molecule, however; it binds to HSA with a high affinity to reach the distant parts of the body. Our results are the first evidence that ATR influences blood parameters together with some metabolites, showing sex-specific pattern. In addition, ATR affects some forms of behavior potentially applicable in the treatment of various diseases.

## Figures and Tables

**Figure 1 life-12-01090-f001:**
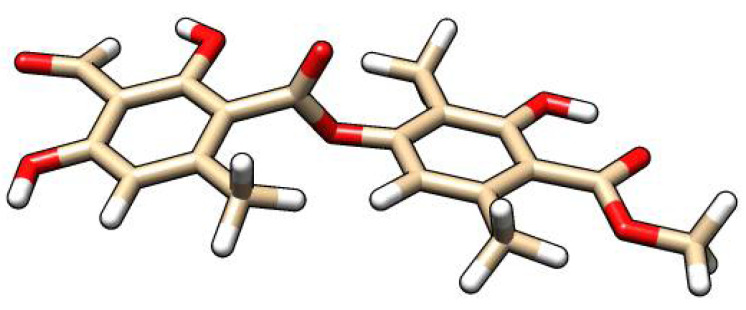
3D model of atranorin structure. The white color indicates an atom of hydrogen (H) and the red color indicates an atom of oxygen (O).

**Figure 2 life-12-01090-f002:**
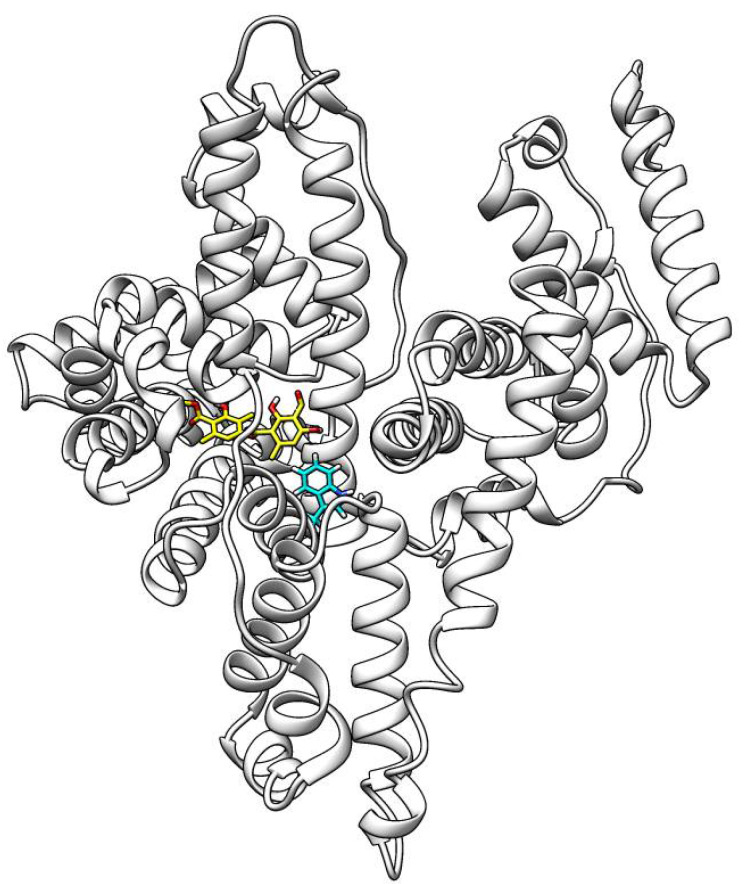
Binding of atranorin near the binding site TRP214 in human serum albumin (data from docking analysis).

**Figure 3 life-12-01090-f003:**
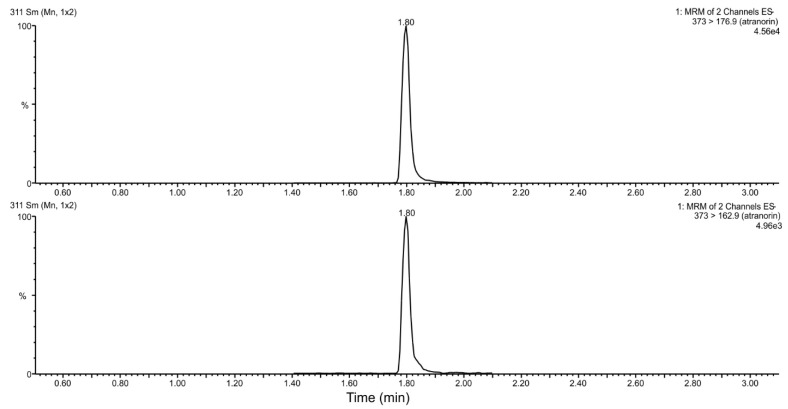
Representative MRM ion-chromatograms of atranorin (20 ng/mL).

**Figure 4 life-12-01090-f004:**
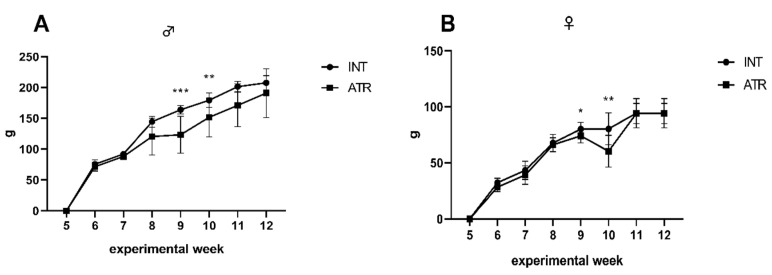
Body mass gain in male (**A**) and female (**B**) animals in INT (healthy intact) and ATR (atranorin) groups. Data are expressed as mean ± standard deviation (SD). Significance vs. INT is given as * *p* < 0.05; ** *p* < 0.01, and *** *p* < 0.001, respectively.

**Figure 5 life-12-01090-f005:**
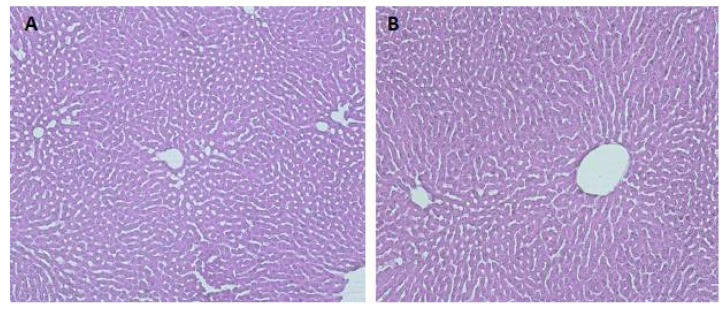
Representative photomicrographs of liver histopathology (200×): (**A**) liver of INT control rats showing normal histology; (**B**) liver of rats after oral administration of 10 mg/kg b.w. of ATR showing normal liver histology.

**Figure 6 life-12-01090-f006:**
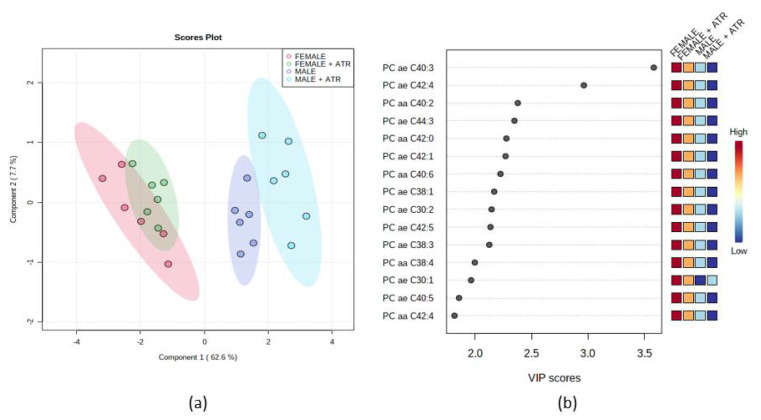
(**a**) Partial least squares-discrimination analysis (PLS-DA) of selected metabolites in ♂INT, ♂ATR, and ♀INT and ♀ATR animals. In the graphical output, 95% confidence ellipses for specific groups are included; (**b**) variable importance in projection (VIP) plot, calculated from PLS-DA method, displays the top 15 most important metabolite features identified by PLS-DA. Boxes on the right indicate the relative concentration of the corresponding metabolite in the blood in descending order of importance. VIP is a weighted sum of squares of the PLS-DA loadings considering the amount of explained Y-variable in each dimension. The most important features have VIP values of >2.0.

**Figure 7 life-12-01090-f007:**
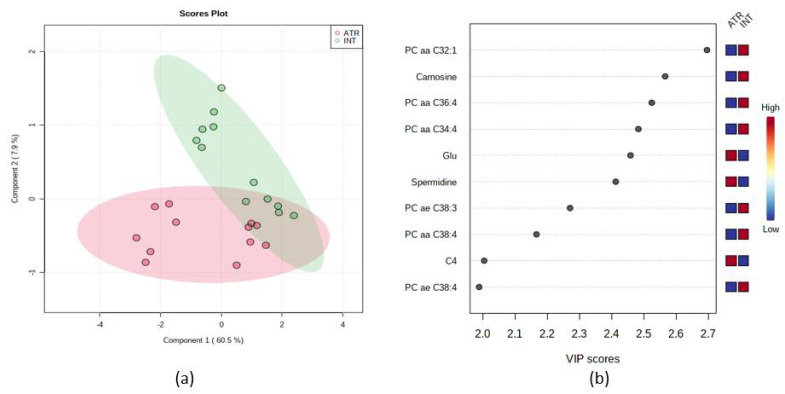
(**a**) Partial least squares-discrimination analysis (PLS-DA) of selected metabolites in INT and ATR animals and intact female. In the graphical output, 95% confidence ellipses for specific groups are included; (**b**) variable importance in projection (VIP) plot, calculated from PLS-DA method, displays the top 10 most important metabolite features identified by PLS-DA. Boxes on the right indicate the relative concentration of the corresponding metabolite in the blood in descending order of importance. VIP is a weighted sum of squares of the PLS-DA loadings considering the amount of explained Y-variable in each dimension. The most important features have VIP values of >2.0.

**Table 1 life-12-01090-t001:** Binding of atranorin (ATR) to human serum albumin (HSA).

	HSA		HSA–ATR	
	1st Peak	2nd Peak	1st Peak	2nd Peak
**peak position** (λex/λem) (nm)	230/335	280/342	230/332	280/336
**relative intensity** (IF)	295	192	171	142
**Δλ**	105	62	102	56

Data are expressed as mean ± SD. HSA—human serum albumin; ATR—atranorin.

**Table 2 life-12-01090-t002:** Calculated quenching and binding parameters of the HSA–ATR system.

Ligand	K_SV_ × 10^5^ (M^−1^)	k_q_ × 10^13^ (M^−1^ s^−1^)	^a^ R^2^	K_B_ × 10^6^ (M^−1^)	n	^a^ R^2^
**HSA**	4.679	4.679	0.977	4.258	1.353	0.997

^a^ R^2^ = correlation coefficient.

**Table 3 life-12-01090-t003:** Food and water intake during the experiment.

		♂		♀	
	Exp. Week	INT	ATR	INT	ATR
**food intake** (g)	**6th**	26.58 ± 2.83	23.75 ± 1.92	17.00 ± 0.73	16.08 ± 1.00
	**11th**	23.00 ± 6.39	21.75 ± 1.73	16.58 ± 3.01	15.75 ± 3.20
**water intake** (mL)	**6th**	46.58 ± 9.40	43.08 ± 3.93	24.83 ± 0.55	25.75 ± 0.28
	**11th**	49.17 ± 5.84	47.75 ± 5.93	37.83 ± 8.40	34.92 ± 7.03

Daily food intake was measured in grams (g) and water intake in milliliters (mL). Data are expressed as mean ± SD. INT—healthy intact, ATR—atranorin.

**Table 4 life-12-01090-t004:** Open field test.

	♂		♀	
	INT	ATR	INT	ATR
**rearing**	9.33 ± 7.57	19.00 ± 8.05 **	12.00 ± 6.00	24.17 ± 3.60 ***
**washing**	5.33 ± 2.80	5.17 ± 2.23	5.17 ± 1.72	4.33 ± 1.03
**defecation**	0.17 ± 0.41	1.17 ± 1.17 *	0.67 ± 0.82	2.17 ± 1.47 **
**time in center**	0.70 ± 0.91	1.11 ± 1.51	0.80 ± 0.72	2.81 ± 3.83
**time on periphery**	360.40 ± 28.76	371.58 ± 6.59	362.22 ± 24.58	374.16 ± 15.48
**traveled distance**	290.88 ± 168.58	266.55 ± 134.95	341.85 ± 136.72	400.65 ± 88.26
**average speed**	0.79 ± 0.43	0.71 ± 0.36	0.95 ± 0.44	0.79 ± 0.29

Data are expressed as mean ± SD. Significance vs. INT is given as * *p* < 0.05, ** *p* < 0.01, and *** *p* < 0.001 respectively. Rearing and washing activities as well as defecation boluses are expressed as counts, time in the center and time on the periphery are expressed in seconds, traveled distance is given in meters and average speed in m·s^−1^.

**Table 5 life-12-01090-t005:** Elevated plus maze.

	♂		♀	
	INT	ATR	INT	ATR
**rearing**	21.33 ± 8.78	22.5 ± 11.98	23.83 ± 10.96	27.17 ± 11.86
**washing**	3.33 ± 1.86	2.83 ± 1.47	2.00 ± 1.41	1.67 ± 1.21
**defecation**	1.17 ± 0.72	2.08 ± 0.51 **	0.00 ± 0.00	1.00 ± 0.89 **
**center crossings**	3.67 ± 3.01	6.00 ± 2.83 *	7.83 ± 3.43	13.17 ± 6.77 *
**time in open arms**	1.17 ± 2.86	35.17 ± 47.20 ***	16.17 ± 15.45	52.83 ± 38.93 *

Data are expressed as mean ± SD. Significance vs. INT is given as * *p* < 0.05, ** *p* < 0.01 and *** *p* < 0.001, respectively. Rearing and washing activities, defecation boluses as well as center crossings are expressed as counts, and time spent in open arms is expressed in seconds.

**Table 6 life-12-01090-t006:** Blood parameters of intact male and female Sprague Dawley rats compared to male and female groups after atranorin administration.

		♂		♀	
		INT	ATR	INT	ATR
**WBC**	10^9^/L	13.10 ± 1.96	15.40 ± 0.83 **	11.57 ± 5.39	8.80 ± 0.75
**LYM**	10^9^/L	10.30 ± 1.36	12.70 ± 1.05 ***	8.87 ± 4.24	6.90 ± 0.53
**MON**	10^9^/L	0.20 ± 0.09	0.23 ± 0.05	0.17 ± 0.13	0.17 ± 0.05
**GRA**	10^9^/L	2.60 ± 0.52	2.47 ± 0.56	2.53 ± 1.20	1.73 ± 0.18
**HCT**	%	43.63 ± 0.95	43.07 ± 1.78	32.90 ± 9.48	34.43 ± 4.33
**HGB**	g/L	161.70 ± 4.77	156.30 ± 8.05	118.70 ± 34.39	126.30 ± 17.02
**RBC**	10^12^/L	8.41 ± 0.28	7.86 ± 0.49 *	5.66 ± 1.77	5.98 ± 0.91
**MCV**	fl	51.95 ± 0.61	54.83 ± 1.38 ***	58.63 ± 1.94	57.80 ± 1.57
**MCH**	pg	19.20 ± 0.31	19.90 ± 0.40 **	21.13 ± 0.61	21.20 ± 0.46
**MCHC**	g/L	370.30 ± 5.22	363.00 ± 4.58	361.00 ± 3.97	366.30 ± 3.91
**RDW**	fl	34.47 ± 0.22	35.57 ± 0.35 ***	35.80 ± 1.31	35.50 ± 0.83
**PLT**	10^9^/L	496.00 ± 62.83	508.00 ± 187.90	112.70 ± 61.90	220.70 ± 119.80
**MPV**	fl	4.90 ± 0.15	4.70 ± 0.31	4.80 ± 0.31	4.97 ± 0.10

Data are expressed as mean ± SD. Significance vs. INT: * *p* ˂ 0.05; ** *p* ˂ 0.01; *** *p* ˂ 0.001. WBC, white blood cells; LYM, lymphocytes; MON, monocytes; GRA, granulocytes; HCT, hematocrit; HGB, hemoglobin; RBC, red blood cells; MCV, mean corpuscular volume; MCH, mean corpuscular hemoglobin; MCHC, mean corpuscular hemoglobin concentration; RDW, red cell distribution width; PLT, platelets; MPV, mean platelet volume; ATR, atranorin.

**Table 7 life-12-01090-t007:** Selected biochemical parameters in healthy male and female Sprague Dawley rats.

		♂		♀	
		INT	ATR	INT	ATR
**LDH**	μkat/L	23.83 ± 6.86	40.07 ± 4.71 ***	17.28 ± 4.46	24.13 ± 3.29 **
**CK**	μkat/L	58.48 ± 32.94	64.68 ± 19.05	25.48 ± 8.27	29.20 ± 13.91
**ALT**	μkat/L	0.91 ± 0.19	1.14 ± 0.50	0.57 ± 0.09	0.53 ± 0.03
**ALP**	μkat/L	2.27 ± 0.23	2.45 ± 0.70	1.42 ± 0.54	1.41 ± 0.25
**T Bil**	μmol/L	0.58 ± 0.50	0.47 ± 0.72	0.52 ± 0.73	0.53 ± 0.58
**Ca**	mmol/L	2.25 ± 0.06	2.30 ± 0.20	2.44 ± 0.06	2.44 ± 0.09
**Crea**	μmol/L	25.53 ± 1.03	32.52 ± 3.53 *	24.30 ± 2.38	28.92 ± 2.84 *
**Urea**	mmol/L	5.78 ± 0.81	5.67 ± 1.06	5.45 ± 0.28	5.48 ± 0.14

Data are expressed as mean ± SD. Significance vs. INT: * *p* ˂ 0.05; ** *p* ˂ 0.01; *** *p* ˂ 0.001. LDH, lactate dehydrogenase; CK, creatine kinase; ALT, alanine amino transferase; ALP, alkaline phosphatase; T Bil, total bilirubin; Ca, calcium; Crea, creatinine; Urea, blood urea; ATR, atranorin.

**Table 8 life-12-01090-t008:** Thiobarbituric acid reactive species (TBARS) content in liver.

	♂	
	INT	ATR
**TBARS** (nmolg/tissue)	42.68 ± 9.06	50.13 ± 11.56
**TBARS content** (nmol)	226.83 ± 39.76	260.04 ± 48.71

Data are expressed as mean ± SD. INT, healthy intacts; ATR, atranorin.

## Data Availability

Not applicable.

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
