# Peer review of "Biochemical Properties of Atranorin-Induced Behavioral and Systematic Changes of Laboratory Rats"

_life, 2022, doi:10.3390/life12071090_

Round 1

Reviewer 1 Report

This study aimed at  basic physicochemical properties, the binding affinity to human serum albumin (HSA), basic pharmacokinetics, and mainly the systematic effects of ATR in vivo. In this study, several tests have been done and it seems suitable for publishing in Life.

1.    Why the collected material was washed in distilled water two times?

2.    Please check the statistical significance of Crea (male) and LDH (female) in Table 7.

3.    This study was performed on rats, not humans so, at the end of the abstract, the sentence should be corrected " ATR at a dose of 10 mg/kg body weight is non-toxic and therefore could become an additive treatment for various diseases."

4.    Please add the aim of the study at the end of the introduction section.

5.    Some of the words should be written italics such as "et al."

6.    The name of section 3.6 is not suitable.

Author Response

Enclosed, please, find the answers to the reviewers, received on 5th July 2022 (life-1818721). We firstly want to thank for your patience, and we thank for all your points to improve our manuscript.

Reviewer 1:

This study aimed at basic physicochemical properties, the binding affinity to human serum albumin (HSA), basic pharmacokinetics, and mainly the systematic effects of ATR in vivo. In this study, several tests have been done and it seems suitable for publishing in Life.

  1. Why the collected material was washed in distilled water two times?

- The material was handled according to previously used method, as was used for example here https://www.ncbi.nlm.nih.gov/pmc/articles/PMC9032407/. The material was washed to get rid of small stones, mold etc. The previous experiences showed that the repeated washing is more effective in removing the small parts of the “dirt”.

  1. Please check the statistical significance of Crea (male) and LDH (female) in Table 7.

- Thank you very much. We checked the statistics and we saw that the copied numbers from the statistical program were not correctly copied. We apologize and we fixed the errors (see in yellow in the manuscript).

  1. This study was performed on rats, not humans so, at the end of the abstract, the sentence should be corrected " ATR at a dose of 10 mg/kg body weight is non-toxic and therefore could become an additive treatment for various diseases."

- Thank you for your point. We rewrote the sentence (yellow). ATR at a dose of 10 mg/kg body weight is non-toxic and therefore could be used in further research.

  1. Please add the aim of the study at the end of the introduction section.

- The aim was added to the end of the section.

  1. Some of the words should be written italics such as "et al."

- Thank you. We fixed it throughout the manuscript.

  1. The name of section 3.6 is not suitable.

- We rewrote the section title. 

Reviewer 2 Report

In the introduction part, Please add some more details, mainly focused on the secondary metabolite atranorin and discussing its effects as an anticancer, I suggesting add some about other parameters. 

In the conclusion part, Please provide some details. 

Author Response

Enclosed, please, find the answers to the reviewers, received on 5th July 2022 (life-1818721). We firstly want to thank for your patience, and we thank for all your points to improve our manuscript.

Reviewer 2:

In the introduction part, Please add some more details, mainly focused on the secondary metabolite atranorin and discussing its effects as an anticancer, I suggesting add some about other parameters.

  • Thank you very much. We added also the immunomodulatory effect of ATR into Introduction (see in yellow).

In the conclusion part, Please provide some details.

  • We added some more information (yellow).

Reviewer 3 Report

The article is suitable for publicating. However, it is the authors' duty to tell the readers what is the problem the authors is trying to solve , why the authors want to solve, and show the meaning of your results. Otherwise I donnot think the article is complete.

Abstract: The authors should explain the purpose or the significance of this project (Why do the authors focus on the topic).  So do the potential effects of your research findings.

Key words: Five words are ok. I donnot think HSA should be a key word.

Inroduction: Your research topic should be introduced in paragraph 3. Research results corresponding to research topic should be introduced in paragraph 4.

Author Response

Enclosed, please, find the answers to the reviewers, received on 5th July 2022 (life-1818721). We firstly want to thank for your patience, and we thank for all your points to improve our manuscript.

Reviewer 3:

The article is suitable for publicating. However, it is the authors' duty to tell the readers what is the problem the authors are trying to solve, why the authors want to solve, and show the meaning of your results. Otherwise I donnot think the article is complete.

  • Thank you. We added the information into the abstract, as well as introduction or conclusion.

Abstract: The authors should explain the purpose or the significance of this project (Why do the authors focus on the topic).  So do the potential effects of your research findings.

  • We rewrote the abstract part so we hope that the purpose of our project is clearer now.

Key words: Five words are ok. I donnot think HSA should be a key word.

  • We do not agree. This is the first evidence that ATR binds to HSA with a high affinity. The docking analysis revealed the position of the binding. That´s why we think it should be listed as the key word.

Introduction: Your research topic should be introduced in paragraph 3. Research results corresponding to research topic should be introduced in paragraph 4.

  • Thank you. We added the aim of the study together with results to the end of Introduction part (see in yellow).